# The autoregulation of a eukaryotic DNA transposon

**Corentin Claeys Bouuaert[1†‡], Karen Lipkow[2,3†], Steven S Andrews[4], Danxu Liu[1], Ronald Chalmers[1]\***

[1]School of Biomedical Sciences, University of Nottingham, Nottingham, United Kingdom; [2]The Babraham Institute, Cambridge, United Kingdom; [3]Department of Biochemistry and Cambridge Systems Biology Centre, University of Cambridge, Cambridge, United Kingdom; [4]Fred Hutchinson Cancer Research Center, Basic Sciences Division, Seattle, United States

**Abstract** How do DNA transposons live in harmony with their hosts? Bacteria provide the only documented mechanisms for autoregulation, but these are incompatible with eukaryotic cell biology. Here we show that autoregulation of Hsmar1 operates during assembly of the transpososome and arises from the multimeric state of the transposase, mediated by a competition for binding sites. We explore the dynamics of a genomic invasion using a computer model, supported by in vitro and in vivo experiments, and show that amplification accelerates at first but then achieves a constant rate. The rate is proportional to the genome size and inversely proportional to transposase expression and its affinity for the transposon ends. Mariner transposons may therefore resist post-transcriptional silencing. Because regulation is an emergent property of the reaction it is resistant to selfish exploitation. The behavior of distantly related eukaryotic transposons is consistent with the same mechanism, which may therefore be widely applicable.

**\*For correspondence:**
chalmers@nottingham.ac.uk

[†]These authors contributed equally to this work

[‡]**Present address:** Molecular Biology Program, Howard Hughes Medical Institute, Memorial Sloan-Kettering Cancer Center, New York, United States

**Competing interests:** The authors declare that no competing interests exist.

**Reviewing editor**: James Ferrell, Stanford University, United States

## Introduction

Transposons are discrete sections of mobile DNA that are amplified as they move from one location in the genome to another. They have had a profound influence on our evolutionary history by providing DNA duplications and rearrangements as substrates for natural selection (e.g. *Hua-Van et al., 2011*). Although retrotransposons, which transpose via an RNA intermediate, can persist for long periods in a given genome, DNA transposons appear to rely on regular horizontal transfer (*Lohe et al., 1995*; *Hartl et al., 1997*). In the period following horizontal transfer the transpositional activity of the element is under positive selection. However, once there are multiple copies in the genome, selection is relaxed because a freely diffusing pool of transposase acts on all copies of the element. Gradually, with time, as various copies of the element acquire detrimental mutations, the pool of active transposase is poisoned by dominant-negative complementation leading to the extinction of the transposon (*Lohe et al., 1995*; *Hartl et al., 1997*; *Le Rouzic and Capy, 2005*; *Le Rouzic et al., 2007*).

Whilst this mechanism likely dictates the ultimate fate of a eukaryotic DNA-transposon, it remains unknown how control is exerted in the short- to medium-term after an active transposon first arrives in a genome. This is a serious problem because transposition is inherently exponential and each new copy of the element is a source of further new copies. The fact that long-lived multicellular organisms tolerate transposons suggests the existence of control mechanisms. Host-mediated epigenetic responses may provide some level of protection. However, these are probably never entirely effective, particularly for euchromatic copies of the transposon (e.g. *Kelleher and Barbash, 2013*). Indeed, modeling suggests that unregulated transposition results either in the demise of the transposon or the demise of the host (*Le Rouzic and Capy, 2005*). Experiments with a non-autonomous Mos1 element in

**eLife digest** Transposons are regions of mobile DNA that can jump from one location in the genome to another. This represents a genetic burden to the host because there is always the risk that the transposon will inactivate a cellular gene. However, a greater problem is that transposition is accompanied by an increase in the number of copies of the transposon. Since each new copy will be a source of further new copies, amplification of transposons is necessarily exponential. The fact that eukaryotic cells are able to tolerate DNA transposons suggests the existence of regulatory mechanisms to defuse the inevitable genomic melt-down. Host-mediated epigenetic modifications and RNA interference will provide some level of protection. However, they are by no means completely effective and a well-adapted genomic parasite, such as a transposon, might be expected to have its own mechanism of regulation.

Now, Claeys Bouuaert, Lipkow and colleagues have used a computer model in combination with in vivo and in vitro experiments to search for this mechanism. Their experiments reveal how a DNA transposon is down-regulated by its own transposase. The transposase is the enzyme that catalyzes the 'jump' or transposition. It binds to specific sites at either end of the transposon and brings these together to make up a nucleoprotein complex called the transpososome. It is within this complex that the chemical steps of the reaction take place. When the number of transposons increases, so does the concentration of transposase. Claeys Bouuaert et al. show that the binding sites become saturated at a relatively low transposase concentration and that negative regulation arises from the resulting competition. Thus, the rate of transposition decreases as the number of transposons increases. They further use the computer model to explore how the amplification of the transposon is affected by transposon-specific and cellular-specific factors.

Claeys Bouuaert, Lipkow and colleagues based their study predominantly on a resurrected copy of the Hsmar1 transposon, which was active in the human genome 50 million years ago. However, they also tested two distantly related eukaryotic transposons and observed that their behavior was similar, which suggests that this could be a general mechanism that controls the activity of jumping genes. They also note that their competition mechanism is conceptually similar to the immunological 'prozone effect'. This is a recurrent theme in protein chemistry and demonstrates once again that less is in fact sometimes more.

*Drosophila* provided the first experimental evidence of autoregulation (*Lohe and Hartl, 1996*). The phenomenon, termed overproduction inhibition (OPI), was revealed by a reduction in the frequency of excision when multiple copies of the transposase gene are present, or when transposase is over-expressed from a heat-shock promoter. OPI was later shown to affect other elements, both in vivo and in vitro (*Lampe et al., 1998*; *Clark et al., 2009*; *Claeys Bouuaert and Chalmers, 2010*).

Our present work focuses on a resurrected copy of the Hsmar1 transposon, which was active in the human genome about 50 million years ago (*Robertson and Zumpano, 1997*; *Cordaux et al., 2006*; *Liu et al., 2007*; *Miskey et al., 2007*). It is closely related to Mos1 and is a member of the mariner family, which is probably the most successful group of transposons in nature, as judged by the depth and breadth of its phylogenetic distribution (e.g. *Robertson and Lampe, 1995*; *Feschotte and Wessler, 2002*). The mariner family is a good model system to address autoregulation because of the demands placed by horizontal transfer on the mechanism of transposition and its control. Ideally, the mechanism should be independent of specific host factors and should allow a rate of transposition sufficiently high to protect the founding element from genetic drift. High activity also provides a high probability of integration into a vector, such as a virus or endosymbiont, which may mediate horizontal transfer. However, this must be balanced against a detrimental effect on host fitness (e.g. *Le Rouzic and Capy, 2005*).

Here we present a mechanism for autoregulation based on a biochemical analysis of Hsmar1 transposition. The model reveals how the kinetics of a genomic invasion are dictated by the multimeric state of the transposase and the order in which the various components are recruited into the developing transpososome. Once a certain number of copies of the transposon are established in the genome, and the transposase concentration is above a critical threshold, the mechanism provides a steady-state rate, perfectly damping the exponential amplification which is a natural consequence of unregulated

transposition. The mechanism is an emergent property of the transposition reaction, based on the competition of active transposase multimers for their primary binding sites at the transposon ends. It is therefore robust and resistant to selfish exploitation. The key prediction of the model is that doubling the transposon copy number, which is tantamount to doubling the transposase concentration, halves the rate of transposition. In vivo transposase dose-response curves for Hsmar1 and the distantly related Sleeping Beauty (SB) and piggyBac (PB) transposons fit this condition, suggesting they are all regulated in the same way.

## Results

### The synapsis-by-protein-dimerization mechanism for transpososome assembly

The documented mechanisms for autoregulation in the bacterial transposons Tn10 and Tn5 provided the starting point for our investigation (*Kleckner, 1990*; *Reznikoff, 2008*). The respective transposases belong to the same family of nucleotidyl-transferases as the Hsmar1 transposase, and use the same cut-and-paste mechanism of transposition (e.g. *Hickman et al., 2010*; see *Figure 2A* below for an illustration of cut-and-paste transposition). The cut-and-paste mechanism does not increase the number of transposons directly. Amplification requires that the empty donor site is repaired by homologous recombination from a sister chromosome, or that the transposon is excised behind a replication fork and integrates in front. However, unless stated otherwise we will assume a maximally efficient reaction in which each transposition event converts one copy of the element into two copies.

In Tn10 and Tn5, the first stage of the reaction is when transposase monomers (T) bind to the ends of the element (A and B) and bring them into a synapsis (*Figure 1A*). This complex is known as a paired ends complex (PEC) or transpososome. We will refer to this mechanism of transpososome assembly as 'synapsis-by-protein-dimerization' (S-PD). A kinetic diagram for the model is provided in *Figure 1B*. The first catalytic steps of the reaction are double strand breaks at the transposon ends, which are irreversible owing to the loss of enthalpy (subsumed into $k_3$ in the diagram). Any regulation must occur before the formation of a productive synapsis because almost all transposons that achieve the first nick go on to complete cleavage and integration (*Chalmers and Kleckner, 1994*, *1996*).

To explore the dynamics of a genomic invasion by a transposon that uses the S-PD mechanism for transpososome assembly we used the kinetic diagram in *Figure 1B* to develop a computer model ('Materials and methods'). At the start of the simulation a single transposon appears in the genome and initiates an exponential amplification (*Figure 1C*). Changing the values of the kinetic constants governing the model changes the scales on the axes but does not alter the exponential character of the reaction. Note that in this case the scale on the X axis is unrealistically short because we allow a productive synapsis to form products almost instantly (the value of $k_3$ in *Figure 1B* is large). We are thus ignoring the time that would be required in vivo for the maturation of the integration products and repair of the donor site. This allows us to focus on the earlier stages of the reaction where regulation takes place.

Amplification of the Tn10 and Tn5 transposons, which use the S-PD mechanism, is controlled by a combination of two effects (*Kleckner, 1990*; *Reznikoff, 2008*). Firstly, the transposases do not diffuse freely and tend to act in cis to the encoding element. This slows the amplification considerably because it prevents the effective concentration in transposase from rising with transposon copy number. However, it does not alter the exponential nature of the curve (*Figure 1D*). Secondly, the transposons express a trans-acting inhibitor, which in the case of Tn10 is an antisense RNA directed against the transposase mRNA. This strategy is also ineffective in dampening the exponential amplification, even if the inhibitor is much more abundant than the transposase (*Figure 1E*). In contrast, the combination of a cis-acting transposase and a trans-acting inhibitor is very effective (*Figure 1F*). This combined strategy is not viable in a eukaryotic host because of the physical separation of transcription and translation. Transposons using the S-PD mechanism would experience exponential amplification in a eukaryote.

### The assembly-site occlusion model for eukaryotic transposons

Whilst the Tn10/5 transposases are monomers in solution, the Hsmar1 enzyme behaves as a dimer in gel filtration experiments (not shown). Furthermore, several mariner transposases readily form a complex known as single end complex 2 (SEC2), which is composed of a transposase dimer bound to a single transposon end (*Lipkow et al., 2004*; *Auge-Gouillou et al., 2005*; CCB & RC data not shown). We

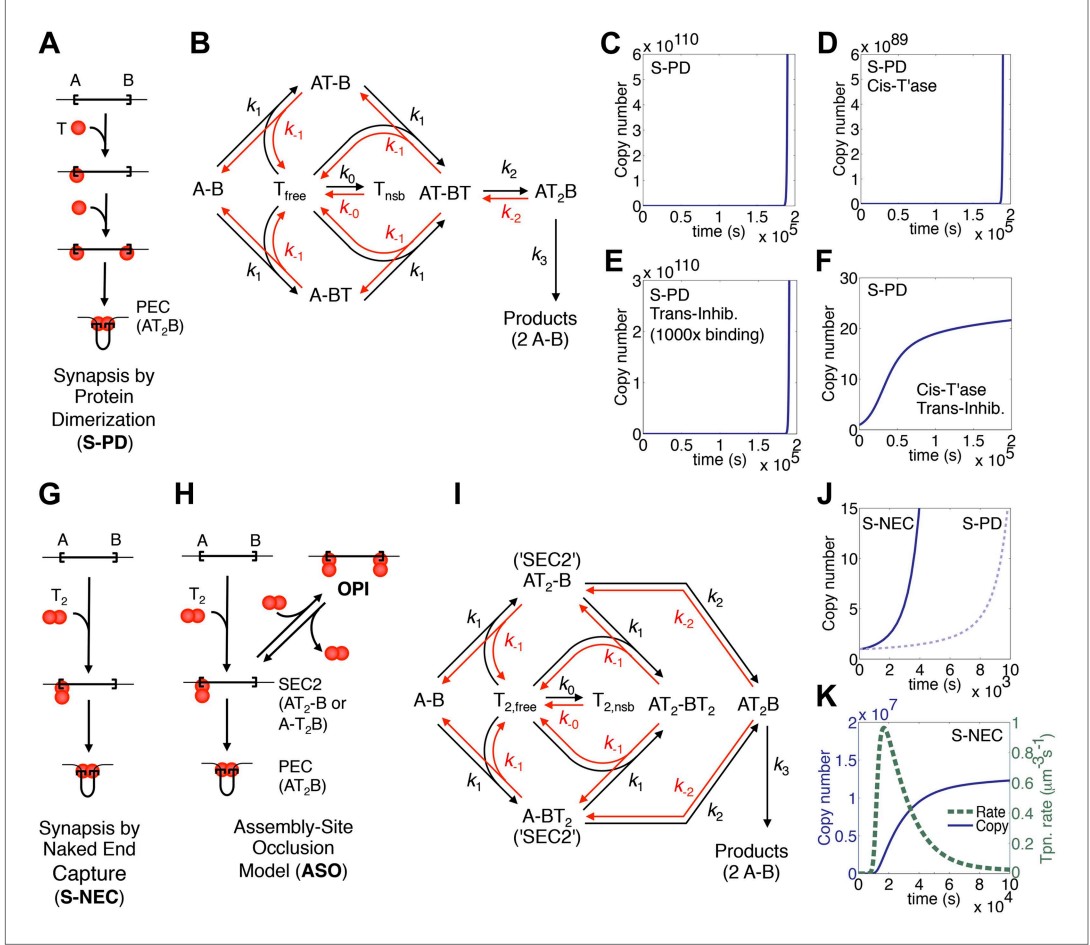

**Figure 1**. Dynamic models for genome invasions. (**A**) The mechanism of transpososome assembly in Tn10/5. In the bacterial elements Tn10 and Tn5 synapsis is mediated by the dimerization of monomers bound to either end of the transposon. T, transposase; $T_2$, transposase dimer; A and B, transposon ends; PEC, paired ends complex. (**B**) The kinetic model embedded in the computer simulation for the S-PD mechanism of synapsis. Abbreviations are as given in part (**A**); $T_{free}$ and $T_{nsb}$, free and non-specifically bound transposase. (**C**) Simulation of the S-PD model. Kinetic parameters: $k_0$, $9.9 \times 10^6$ M$^{-1}$s$^{-1}$; $k_{-0}$, 139 s$^{-1}$; $k_1$, $3.8 \times 10^8$ M$^{-1}$s$^{-1}$; $k_{-1}$, $1.2 \times 10^{-2}$ s$^{-1}$; $k_2$, 12.7 s$^{-1}$; $k_{-2}$, $1 \times 10^{-10}$ s$^{-1}$; $k_3$, $1.4 \times 10^{-3}$ s$^{-1}$. Sources of the values are given in **Table 2**. (**D**) As in part (**C**) but the transposase is 99% cis-acting. 1% of the transposase leaks into the bulk phase and acts on all copies of the element. (**E**) As in part (**C**) but the transposon expresses a trans-acting inhibitor which reduces the transposase concentration. The inhibitor is 1000-fold more active than the transposase, which approximates the situation in Tn10 where the inhibitor is an antisense RNA. (**F**) The cis-acting transposase and the trans-acting inhibitor from parts (**D**) and (**E**) are combined. (**G**) The mechanism of transpososome assembly in mariner. A transposase dimer bound to one transposon end recruits a naked transposon end. Abbreviations as in part (**A**). (**H**) The assembly-site occlusion model. When the transposase concentration is low, most transposons are occupied by only one transposase dimer, leading to productive synapsis (bottom left). When the transposase concentration is high, a transposon may be occupied by two transposase dimers, causing overproduction inhibition (OPI, top right). SEC2 is a transposon with a transposase dimer bound to one of the ends. (**I**) Kinetic model embedded in the computer simulation for the ASO mechanism. Abbreviations as given in parts (**A**) and (**B**). (**J**) Simulation of the S-NEC mechanism and comparison with S-PD. Parameters as in part (**C**). (**K**) The time scale of the S-NEC simulation in part (**J**) is extended.

would like to propose that SEC2 gives rise to the transpososome by capturing a naked transposon end (**Figure 1G**). We will refer to this mechanism as synapsis-by-naked-end-capture (S-NEC). Recruitment of a naked transposon end immediately suggests a mechanism that can account for OPI (**Figure 1H**). As the concentration of transposase rises, transposition is inhibited by the progressive reduction in the number of unbound ends available for synapsis. We will refer to this as an assembly-site-occlusion model

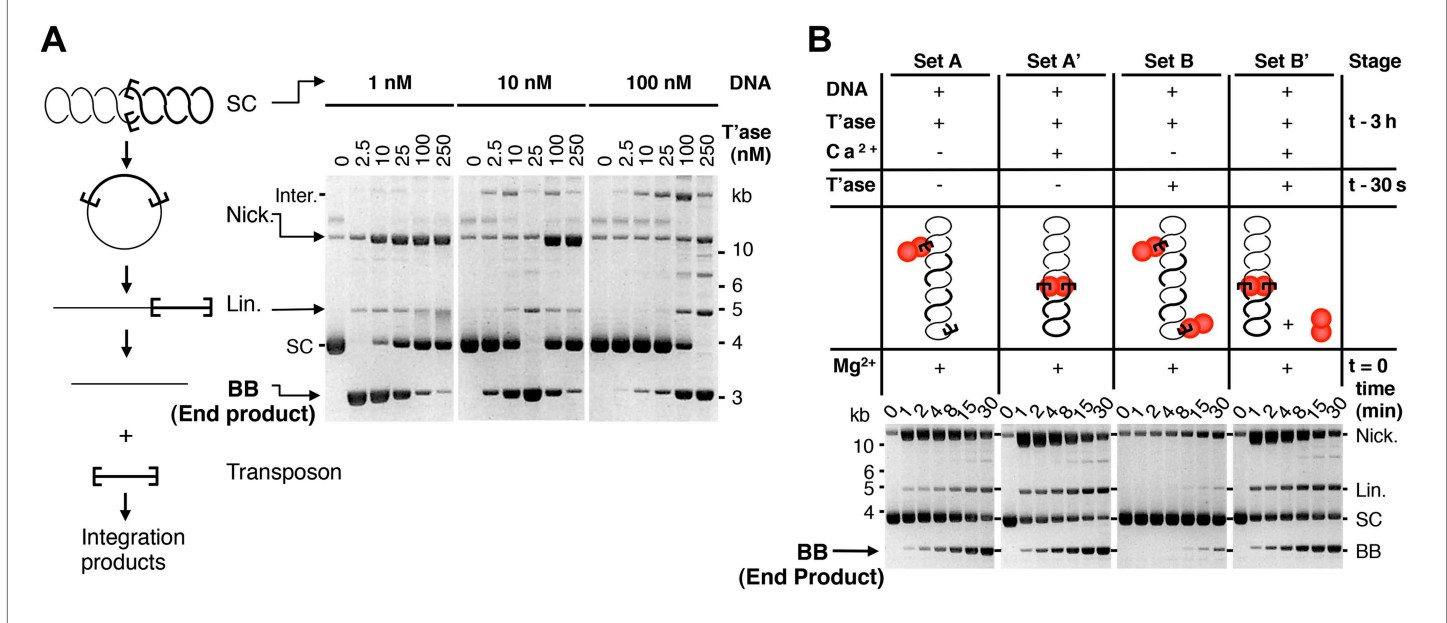

**Figure 2**. Experimental test of the ASO model for mariner transposition. (**A**) The efficiency of transposition depends on the ratio of transposase to DNA. The cartoon to the left of the gels illustrates the mechanism of cut-and-paste transposition using a supercoiled (SC) substrate. First strand nicking generates an open circular product (Nick.). Second strand nicking at one end yields the linear single-end break product (Lin.). A similar set of nicks at the other transposon end yield the plasmid backbone (BB) plus the excised transposon. Transposase was titrated into reactions with three different substrate concentrations. Reactions contained 1.5 µg of supercoiled substrate plasmid in a volume of 500 µl, 50 µl and 5 µl, which provided a final concentration of 1 nM, 10 nM and 100 nM, respectively. Transposase was diluted so that the addition of one tenth of the respective reaction volumes achieved the indicated concentrations. Reactions were incubated for 4 hr at 37 °C and deproteinated. Photographs of ethidium bromide stained agarose gels are shown. Consumption of the supercoiled substrate and production of the plasmid backbone both provide a measure of the efficiency of the reaction. Inter., product of intermolecular integration of the transposon into an unreacted substrate. (**B**) Preassembly of the paired-end complex protects from OPI. Four sets of staged reactions were assembled as indicated. Set A was a standard transposition reaction with 6.7 nM plasmid substrate and 20 nM of transposase except that the components were pre-incubated for 3 hr before the addition of the catalytic $Mg^{2+}$ ion. Set A' was identical except that $Ca^{2+}$ was present during the 3 hr pre-incubation period, which enhances PEC assembly. Sets B and B' were identical to A and A' except that the mixture was supplemented with 50 nM transposase 30 s before the addition of $Mg^{2+}$. OPI inhibited the reaction in Set B. In contrast, the stable PEC assembly supported by the presence of $Ca^{2+}$ protected Set B' from the inhibitory effects of the excess transposase added just before the catalytic $Mg^{2+}$ ion. Photographs of ethidium bromide stained agarose gels are shown. See also *Figure 2—figure supplement 1*.

The following figure supplements are available for figure 2:

**Figure supplement 1**. Kinetic analyses of OPI and interpretation of in vitro transposition reactions.

(ASO). A kinetic diagram for the model is provided in *Figure 1I*, following the same conventions as before, except that $T_2$ represents the transposase dimer.

To explore the dynamics of mariner transposition, we simulated the S-NEC mechanism of transpososome assembly in a second computer model ('Materials and methods'). Using an identical set of parameters, S-NEC has an advantage over the S-PD mechanism in terms of the rate at early time points (*Figure 1J*). However, as the invasion progresses, the S-NEC model is dampened by ASO, which eventually dominates the reaction (*Figure 1K*). The peak rate of the reaction is when the free-transposase concentration is equal to its equilibrium binding constant.

## Verification of the ASO model

The ASO model provides testable predictions. The simplest is that the rate of transposition is governed by the ratio of transposase to transposon ends, and not by the absolute concentrations. We tested this using the in vitro excision assay (*Figure 2A*, but please also consult *Figure 2—figure supplement 1* for a guide to interpretation of the in vitro reactions). Excision of the transposon leaves behind the vector backbone, which is an end product of the reaction and provides a direct measure of the efficiency. With 10 nM substrate, the reaction peaked at 25 nM transposase and then declined sharply, as judged by the amount of backbone produced. When the concentration of substrate was

increased or decreased by 10-fold, the amount of transposase required for peak activity changed in direct proportion. This is consistent with the ASO model, and excludes models in which OPI arises from a concentration-dependent mechanism of transposase aggregation.

Another prediction is that pre-assembly of the PEC will sequester the transposon ends in productive interactions, and protect the reaction from the subsequent addition of excess transposase. To test this hypothesis we measured the kinetics of four transposition reactions, which we assembled in different stages prior to the addition of the catalytic $Mg^{2+}$ ions at time zero (*Figure 2B*). Set A is a standard transposition reaction in which the components were incubated for 3 hr prior to the addition of $Mg^{2+}$. Set A' is identical except that $Ca^{2+}$ was present during the pre-incubation period. Although $Ca^{2+}$ does not support DNA cleavage, it supports assembly of the PEC (*Claeys Bouuaert and Chalmers, 2010*). In Set A' this is evident from the faster consumption of the supercoiled substrate. Upon addition of $Mg^{2+}$, 50% of the substrate was converted from the supercoiled to the nicked form within 1 min.

Set B is identical to Set A, except that extra transposase was added 30 s before the catalytic metal ion at time zero (*Figure 2B*). As expected, this inhibited the reaction and very little substrate was converted into product. Set B' is identical to Set B except that $Ca^{2+}$ was present during the pre-incubation period, before the addition of excess transposase. The reaction kinetics were then identical to those of Set A'. Pre-assembly of the transpososome is completely effective in the relief of OPI because the competition for free transposon ends is irrelevant at this stage of the reaction. Transposition is thus regulated during assembly of the transpososome, before productive synapsis of the ends.

## Synapsis is very sensitive to the rate of diffusion

We have assumed that each copy of the transposon contributes a certain amount of transposase to the nucleus and that this is where transposition takes place. We therefore estimated the rate of diffusion in this compartment and calculated its effects on the association rate constants for specific and non-specific DNA ('Materials and methods'). Compared to the in vitro rates, the values were reduced by about twofold, but their effects cancel each other out and the rate of transposition is unchanged as a result. However, the effect of diffusion on the rate of synapsis is much greater because of the drag experienced by a chromosomally-bound protein. By treating the DNA with the worm-like chain model we obtained an estimate for 'segmental-diffusion', which describes the rate of motion on a short scale, such as the distance between the transposon ends ('Materials and methods'). Our calculated value is consistent with the experimentally determined rate of looping derived from lox/Cre recombination in mammalian cells (*Ringrose et al., 1999*). When the calculated diffusion rates are applied in the computer model, a pseudo-steady-state rate of transposition is established when there are about 200 copies of the transposon (*Figure 3A*, compare with *Figure 1K*). This steady-state rate persists well beyond the copy numbers typically achieved during a genomic invasion (*Figure 3—figure supplement 1*).

## Hsmar1-specific factors

Up until now we have assumed that synapsis of the transposon ends is the product of two sequential, chemically identical, collision events: a transposase dimer collides with and binds first one transposon end and then the other. This system is useful to illustrate the fundamental dynamics of a generic DNA-looping reaction, and how it is governed by the concentration of the looping protein. However, the Hsmar1 transpososome is more complicated because of communication between the transposase subunits. The allostery was first noted in competition experiments using different topological forms of the substrate (*Claeys Bouuaert et al., 2011*). The experiments revealed that transposase binds quickly and tightly to the first transposon end it encounters, but has a much lower affinity for the second transposon end (*Claeys Bouuaert et al., 2011*). To see how this affects the rate of transposition we estimated the association and dissociation rate constants, and the rate of synapsis, in in vitro experiments. To estimate the association rate constants ($k_1$ in *Figure 1I*) we used an electrophoretic mobility shift assay (EMSA). Assembly of SEC2 was complete after 30 s, which was the minimum time required to load the gel and apply the voltage (*Figure 3B*). Although this did not provide a direct measure of $k_1$, it was consistent with the rapid site-specific binding of a helix-turn-helix protein and we therefore retained the previous best estimate from the literature ('Materials and methods').

To estimate the dissociation rate constant ($k_{-1}$ in *Figure 1I*) we challenged SEC2 with a 10-fold molar excess of unlabeled transposon end (*Figure 3C*). The complex dissociated slowly and about 50% remained after 20 min. We therefore estimate that $k_{-1}$ is $5.8 \times 10^{-4}$ $s^{-1}$. This is some 20-fold lower than the value for most helix-turn-helix proteins ('Materials and methods'). It is possible that the slow

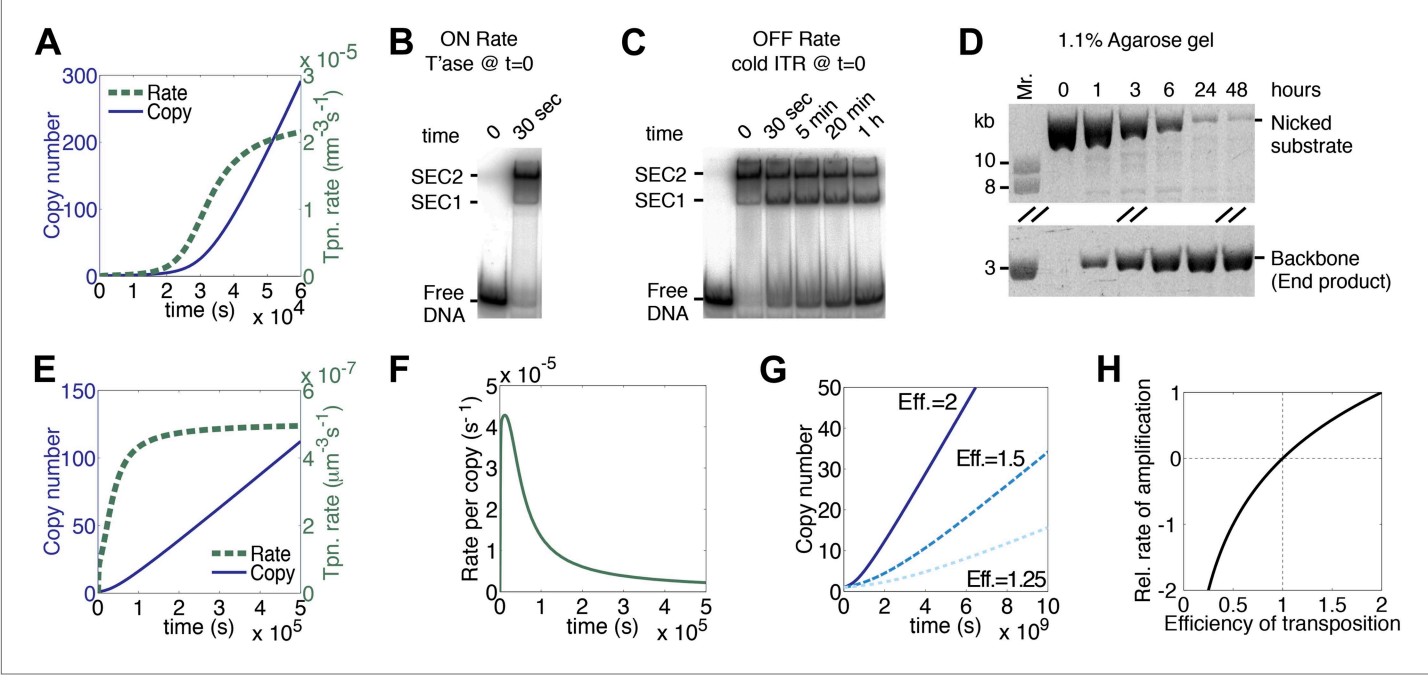

**Figure 3**. A semi-quantitative description of mariner transposition. (**A**) The ASO model for mariner transposition was simulated after taking account of the slow diffusion which prevails in vivo: $k_0$, $5 \times 10^6$ M$^{-1}$s$^{-1}$; $k_1$, $1.9 \times 10^8$ M$^{-1}$s$^{-1}$; $k_2$, $4.5 \times 10^{-4}$ s$^{-1}$. Other parameters were as in **Figure 1K**. (**B**) Transposase binds the transposon end rapidly and tightly. Binding reactions (20 µl) contained 40 fmol of $^{32}$P-labeled transposon end and 120 fmol of transposase. The reactions were incubated at 37 °C and separated on a native polyacrylamide gel. The lane indicated as time zero contains no transposase. The 30 s time interval was the time required to mix the sample, load the gel and apply the voltage. SEC1 and SEC2 represent a single transposon end bound by a transposase monomer and dimer, respectively. An autoradiogram is shown. (**C**) Binding reactions were set up and analyzed as in part (**B**). After 5 min incubation, to allow the complexes to form, a 20-fold molar excess of unlabeled transposon end ('cold ITR') was added. The lane indicated as time zero contains transposase but no cold competitor. The rate of transposase dissociation can be estimated from the amount of free DNA released from the complexes. (**D**) The rate of transposon excision from a nicked substrate provides an estimate of the rate of synapsis. The kinetics of a transposition reaction was analyzed in standard reactions containing 6.7 nM of nicked plasmid substrate and 20 nM transposase. A photograph of an ethidium bromide stained agarose gels is shown. (**E**) The ASO model was simulated as in **Figure 1K** but taking account of the effects of allostery, as described in the main text and 'Materials and methods'. Parameters as in **Figure 1K** except $k_2 = 9.6 \times 10^{-5}$ s$^{-1}$; $k_{-1} = 5.8 \times 10^4$ s$^{-1}$. (**F**) As in part (**E**) but the rate of transposition is divided by the transposon copy number. (**G**) As in part (**A**), but taking account of allostery and slow diffusion in vivo: $k_0$, $5.0 \times 10^6$ M$^{-1}$s$^{-1}$; $k_1$, $1.9 \times 10^8$ M$^{-1}$s$^{-1}$; $k_2$, $3.4 \times 10^{-9}$ s$^{-1}$. The effects of changing transposition efficiency are shown (see text for definition of efficiency). (**H**) The relationship between transposition efficiency and the relative rate of transposition is plotted with the maximum rate scaled to 1: $y = 1.4427 \ln(x)$. See also **Figure 3—figure supplements 1 and 2**.

The following figure supplements are available for figure 3:

**Figure supplement 1**. The time scale of the graph in **Figure 3A** is extended.

**Figure supplement 2**. EMSA analysis of transpososome assembly shows that SEC1 arises from dissociation of the PEC.

dissociation reflects a second aspect of the aforementioned allostery. In this experiment we also observed single-end-complex 1 (SEC1), which contains a monomer of transposase bound to a single transposon end. It arises from decay of the PEC, which is unstable in the EMSA (**Figure 3—figure supplement 2A–C**). SEC1 is highly abundant in this experiment because the excess of cold competitor ends drives PEC assembly by mass action (**Figure 3—figure supplement 2D**).

To measure the rate of synapsis ($k_2$) we used a transposon encoded on an open circular plasmid (**Figure 3D**), which is probably the most relevant substrate because eukaryotic DNA contains very little free supercoiling. The half-time for synapsis was estimated at 2 hr, which corresponds to a pseudo-first order rate constant of $9.6 \times 10^{-5}$ s$^{-1}$ ('Materials and methods').

When the experimental estimates of $k_{-1}$ and $k_2$ were applied in the computer model for the generic DNA looping reaction in vitro, the rate of transposition was greatly reduced and the accelerating phase at the beginning of the reaction was largely eliminated (**Figure 3E**, compare with **Figures 1K**

*and 3A*). Although the overall rate of transposition remains constant, the rate for any particular copy of the element decays exponentially (*Figure 3F*). Finally, when we also take account of the slow diffusion in vivo, the rate of transposition is reduced even further (*Figure 3G*, solid line).

## The efficiency of transposition in vivo

Cut-and-paste transposition relies on the host homologous recombination machinery to reinstate a copy of the transposon at the donor site following excision. The highest rate of amplification, which we define as an efficiency of two, is achieved if, furthermore, the transposon excises after passage of a replication fork and inserts into an un-replicated region ahead of the fork. In this situation, one element on one chromosome becomes four elements on two sister chromosomes. The rate of transposition is quite sensitive to the efficiency of the reaction (*Figure 3G,H*). If the transposon inserts into a replicated region of the chromosome, or if the donor site is not reinstated, or if the excised transposon fails to reintegrate, the efficiency is reduced to 1.5 and the rate of transposition is halved. The rate of transposition may also be negative (*Figure 3H*).

## Association and dissociation rate constants

The computer model reveals an inverse linear relationship between the association rate constant ($k_1$) and the rate of transposition (*Figure 4A*). Increasing the affinity of the transposase for the transposon end reduces the rate of transposition, and vice versa. This is because the transposase acts primarily as an inhibitor of its own activity once the steady-state is established. In principle, this behavior could cause the preferential amplification of transposons which had acquired mutations in the transposase binding sites. However, when we reduced the association rate constant by 100-fold it was clear that the advantage was accompanied by a penalty in the form of a lag phase at the start of the invasion (*Figure 4B*). A low-affinity founding element would therefore be vulnerable to genetic drift during the period before sufficient copies had been produced to sustain the transposase concentration required for efficient amplification.

To verify this property of the model we mutagenized the transposase to lower its affinity for the transposon end. The RA104 mutation is located within the second helix-turn-helix motif, and abolishes an ionic interaction with the phosphodiester backbone (*Roman et al., 2007*; *Richardson et al., 2009*). In the EMSA the RA104 transposase produced very little SEC2, even when the standard transposase concentration was increased threefold (*Figure 4C*, compare with *Figure 3B*). However, in the in vitro plasmid transposition assay, RA104 consumed the supercoiled substrate faster than wild type (*Figure 4D*). Presumably, the mutant protein redistributes more quickly from those plasmids initially suffering OPI due to the double occupancy of their transposon ends. When the transposase concentration was increased 10-fold the wild type activity was almost abolished, whereas the mutant was almost unaffected (*Figure 4D*).

We next turned to an established eukaryotic cell culture assay. It is based on the co-transfection of a helper plasmid, expressing transposase, and a reporter plasmid, encoding a transposon that confers neomycin resistance when it integrates into a host chromosome (e.g. [*Grabundzija et al., 2010*]). The rate of transposition is given by the number of stable transfectants obtained after drug selection. With the optimum amount of helper plasmid RA104 was only 25% as active as wild type. However, with higher levels RA104 was less severely inhibited (*Figure 4E*). Thus, the RA104 transposase reflects the behavior of the low affinity element in *Figure 4B*, which is at a disadvantage to wild type when the transposase concentration is low.

## Transposase expression level

Once a transposon is established in the genome, ASO provides a constant rate of transposition. With a given set of kinetic parameters for transposase binding and synapsis, the actual rate achieved depends on the amount of transposase expressed by each copy of the transposon (*Figure 5A*). Reducing the expression level by fivefold increases the steady-state rate by the same factor. However, it takes slightly longer to establish the final rate because of the difficulty locating the transposon ends at the start of the invasion when the transposase concentration is low. The limits of this effect are reached when the expression of transposase dimers is fractionally sub-stoichiometric to transposon ends (*Figure 5B*). Since ASO requires at least two dimers per transposon, the accelerating phase is never dampened and amplification, although slow at first, is exponential.

We next considered a genome with 1000 copies of the transposon and calculated the rate of transposition over a range of transposase expression levels (*Figure 5C*). This revealed an inverse-exponential

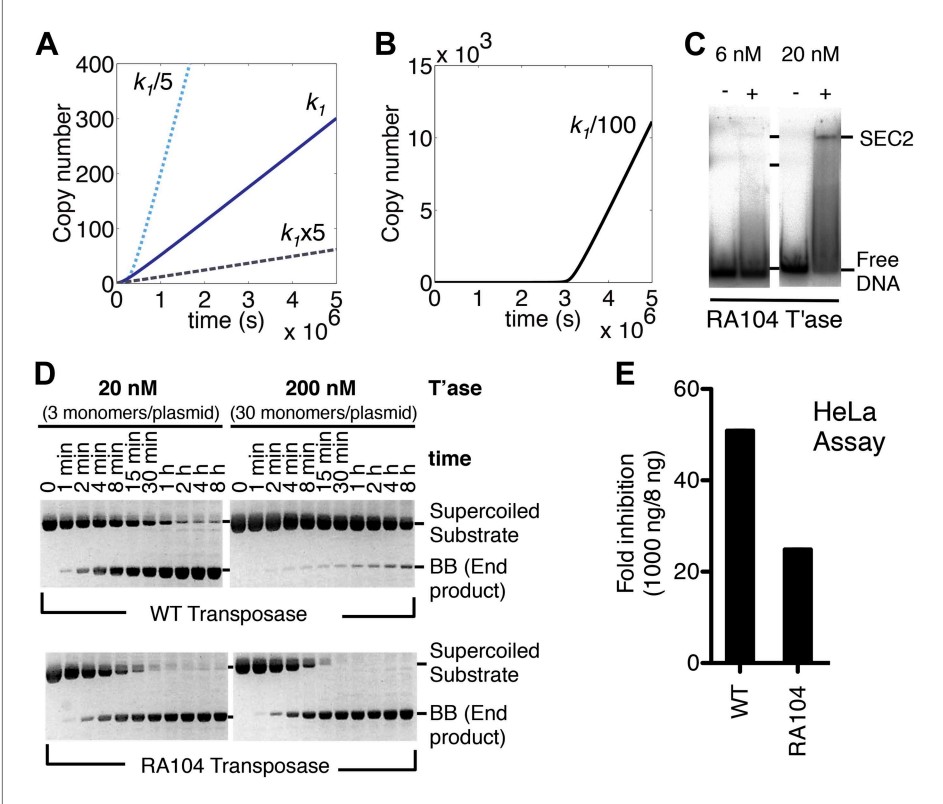

**Figure 4**. Association and dissociation rate constants. (**A**) Simulation as in *Figure 3E*, with $k_1$ fivefold up or down. We have retained the effects of allostery but here and in subsequent simulations we have ignored the effects of the slow diffusion in vivo. This allows the algorithm to run more smoothly and shortens the scale on the x axis, but does not affect the conclusions, which are based on the differential responses to changing the various parameters. (**B**) Simulation as in part (**A**) (solid line), with $k_1$ 100-fold down. (**C**) Binding reactions with the RA104 mutant transposase were set up as in *Figure 3B*. In the lane with 20 nM transposase the smear between SEC2 and the position of free DNA is probably due to complexes that have dissociated during electrophoresis. Autoradiogram of an EMSA is shown. (**D**) The kinetics of the transposition reaction were analyzed in standard reactions containing 6.7 nM of supercoiled plasmid substrate and the indicated concentrations of the transposases. Photographs of ethidium bromide stained agarose gels are shown. With 200 nM wild-type transposase, the windows of opportunity for synapsis, provided by periods when one transposon end is unoccupied, are too short to allow for synapsis. The RA104 transposase mutant is resistant to OPI because the higher dissociating rate provides more windows of opportunity for synapsis. (**E**) Mutant and wild type transposase were assayed in HeLa cell culture with 8 ng or 1000 ng of helper plasmid and 500 ng of neomycin resistant reporter plasmid.

relationship between transposase expression and the rate of transposition. This is noteworthy because it suggests that once the steady-state is established, a post-transcriptional silencing response may potentiate the genomic invasion. It also suggests that titration of the transposase by non-autonomous elements may not contribute to the demise of a transposon as has been previously suggested.

To verify the inverse relationship between the transposase concentration and the rate of transposition, we titrated the helper plasmid in the HeLa cell culture assay (*Figure 5D*). The inhibitory response followed the power law relationship predicted from the ASO mechanism ($y = ax^{-1}$). A Western blot was used to confirm that transposase expression increased with increasing amounts of helper plasmid (*Figure 5—figure supplement 1*). We also examined the transposase dose-response relationships for two other eukaryotic DNA transposons (*Figure 5E,F*). Sleeping Beauty (SB) is distantly related to Hsmar1 within the mariner/Tc1 superfamily (*Goodier and Davidson, 1994*; *Ivics et al., 1997*). The lepidopteran transposon piggyBac is very distant from Hsmar1

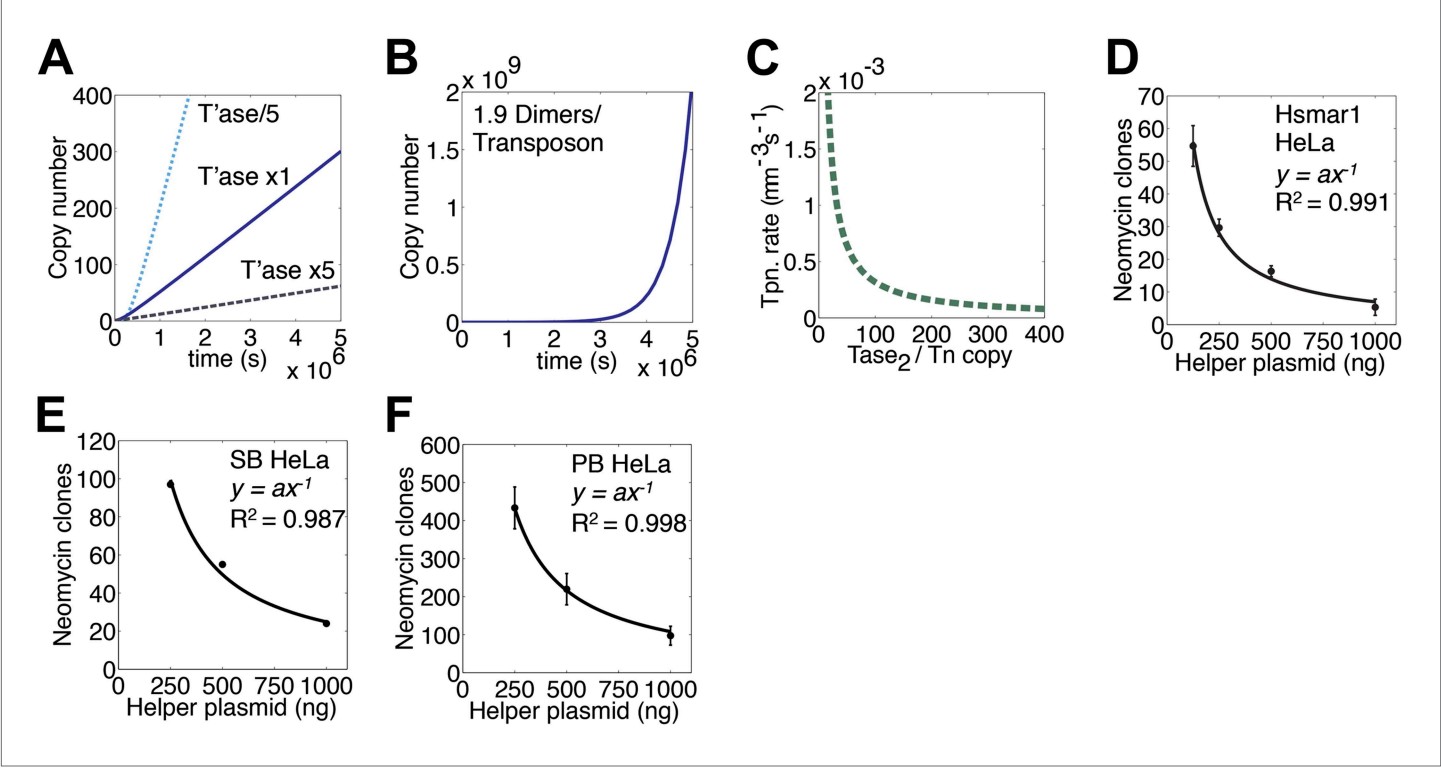

**Figure 5**. Transposase expression level. (**A**) Simulation as in *Figure 3E* showing the effects of changing the transposase expression fivefold up or down. Unless noted otherwise, each transposon copy produces 500 transposase dimers, which are considered as being contained within a 500 fl nucleus. (**B**) As in part (**A**) (1× line) but with a transposase expression level of 1.9 dimers per transposon. (**C**) Parameters as part (**A**) (1× line), plotting the relationship between transposase expression and the rate of transposition at the point in the invasion when there are 1000 copies of the transposon present in the genome. (**D**) HeLa cell assay for Hsmar1 transposition. HeLa cells were transfected with 500 ng of neomycin reporter plasmid and the indicated amount of helper plasmid expressing transposase. The rate of transposition is given by the number of neomycin resistant colonies recovered after drug selection. Bars indicate standard error of the mean, n = 3. $R^2$ is least squares goodness of fit to the line $y = ax^{-1}$. (**E**) As in part (**D**) but with isogenic Sleeping Beauty (SB100X) transposon reporter and helper plasmids. Data points are a mean of three experiments and were extracted and re-plotted from *Figure 2A* of (*Grabundzija et al., 2010*). (**F**) As in part (**D**) but with isogenic piggyBac transposon reporter and helper plasmids. n = 3. See also *Figure 5—figure supplement 1*.

The following figure supplements are available for figure 5:

**Figure supplement 1**. Transposase expression in HeLa cells.

(*Fraser et al., 1996*). In both cases the transposase dose-response curve matched the predicted power law relationship.

## Genome size and nuclear volume

Mariner transposons are widely distributed in plants and animals where genome sizes and nuclear volumes vary greatly. In the S-NEC computer model, the rate of transposition is barely sensitive to volume changes less than 100-fold (*Figure 6A*). This is because changes in the amount of transposase bound to specific and non-specific sites almost exactly cancel each other. The S-PD mechanism of synapsis responds in the opposite direction (*Figure 6B*).

Increasing the size of the genome in the S-NEC/ASO model increases the rate of transposition in direct proportion (*Figure 6C*). This is a result of the inverse relationship between the transposase concentration and the rate of transposition, which was verified in *Figure 5D*. The effect arises from the facts that the number of non-specific binding sites scales with the genomes size, and that these relieve OPI by absorbing free transposase. Whether or not non-specific sites are composed of DNA or chromatin does not matter. The ASO mechanism thus tailors the rate of transposition according to the genome size. The S-PD model responds in the opposite direction, and smaller genomes receive progressively higher rates of transposition (*Figure 6D*).

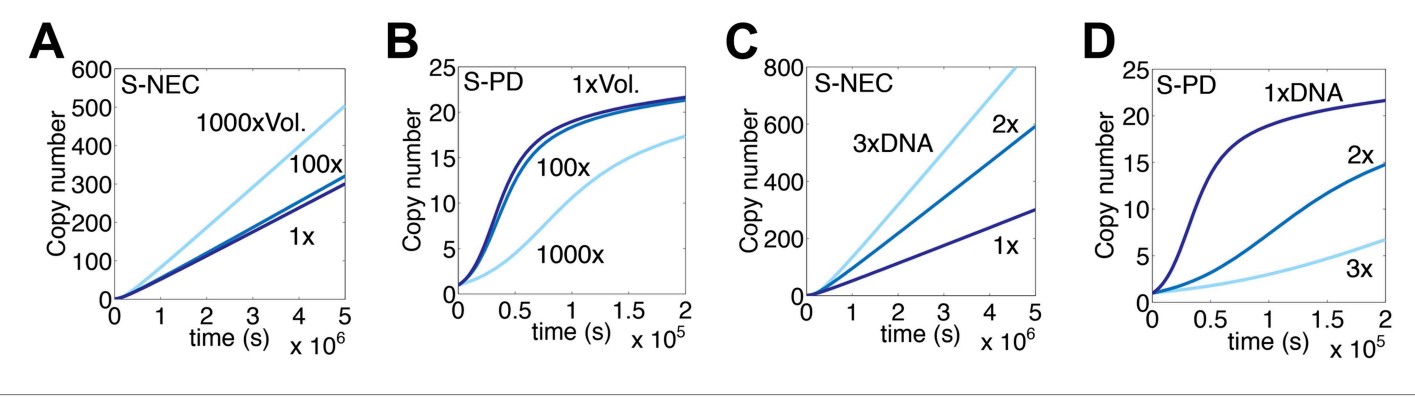

**Figure 6**. Genome size and nuclear volume. (**A**) Simulation as in **Figure 3E**, changing the nuclear volume by the indicated amounts. (**B**) Simulation of regulated S-PD mechanism as in **Figure 1F**, changing the nuclear volume by the indicated amounts. (**C**) Simulation as in part (**A**), changing the genome size by the indicated amounts. (**D**) Simulation as in part (**B**), changing the genome size by the indicated amounts.

## Variations and alternative models

The rice transposon Osmar14 has a secondary transposase binding site close to one of the two principal binding sites at the transposon ends (**Yang et al., 2009**). It was suggested that transposase binding at the secondary site may down-regulate transposition. To investigate the behavior of such a system, we introduced an inhibitory binding site into the S-NEC reaction scheme (**Figure 7A**). The steady-state rate of transposition was reduced in proportion to the binding affinity of the transposase. Note that the behavior of the system would be identical if the transposon instead expressed a transposase variant or a completely different protein, provided that binding interfered with transposition.

In the context of the S-PD reaction scheme, a secondary transposase binding site is identical to the trans-acting inhibitor modeled in **Figure 1E**. Thus, even if the inhibitory binding site is stronger than the principal binding sites, it will not halt the exponential amplification, but merely slow it down. This highlights the weakness of 'secondary site' models for regulation in the absence of the dose-dependent inhibition provided by the ASO mechanism.

ASO is reminiscent of the dimerization end-occlusion model (DEO) first proposed for Tn5 and later in more detail for the Mos1 mariner transposon (**Weinreich et al., 1994**; **Townsend and Hartl, 2000**). The model postulates an S-PD mechanism of transpososome assembly, accompanied by the concentration-dependent formation of inactive dimers, which bind and occlude the transposon ends (**Figure 7B**). Tn5 was later shown to be regulated by other factors (above), and the current biochemical analysis excludes the DEO mechanism for mariner transposition. Nevertheless, DEO is a viable model and we wished to explore its properties: firstly to compare them with ASO, and, secondly, to facilitate its recognition should it be present in any natural element that may be characterized in the future.

All potential states of the DEO model are illustrated in **Figure 7C**. However, in constructing a computer model we used a slightly simplified version in which only free monomers produce inactive dimers. In this scheme there is a single potentially active species, shown in red. As previously demonstrated, the model provides the dose-dependent inhibition needed to prevent exponential amplification (**Townsend and Hartl, 2000**; **Figure 7D,E**). The eventual steady-state rate is inversely proportional to the stability of the inhibitory dimer (**Figure 7E**). As in the ASO model, the steady-state rate is also proportional to the genome size (**Figure 7F**). However, larger genomes experience a longer lag phase at the start of the invasion. This is an inherent disadvantage of the S-PD mechanism of transpososome assembly and reflects the improbability of transposase monomers binding to both ends of the element simultaneously when their concentration is low (**Figure 1J**). Interestingly, once established, the steady state rate is independent of the association and dissociation rate constants, $k_1$ and $k_{-1}$, provided that dimerization does not change either of these values (**Figure 7G** and not shown).

## Discussion

DNA transposons require regulation to balance their fitness at different stages of a genomic invasion (**Le Rouzic and Capy, 2005**). Initially, a high rate of transposition is desirable as it guards against

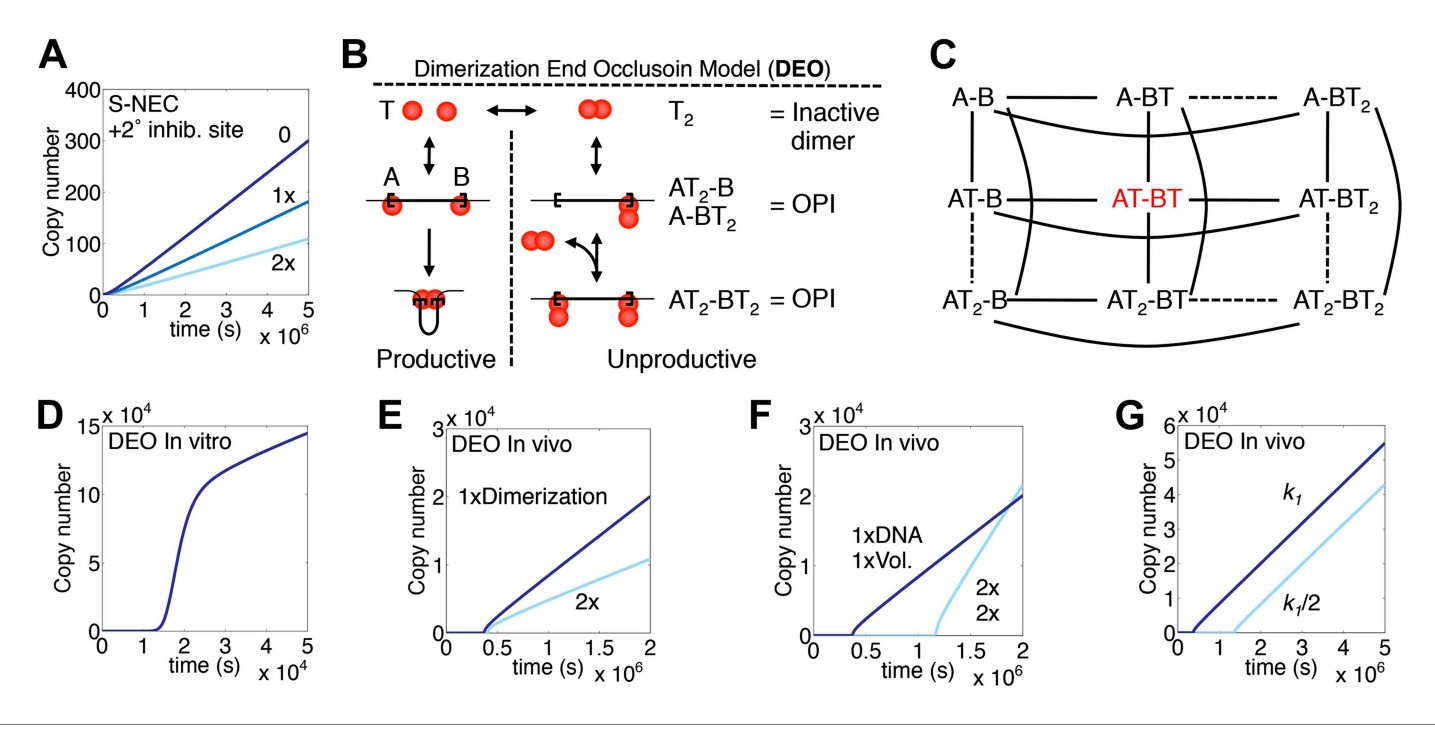

**Figure 7**. Variations and alternative mechanism of regulation. (**A**) The S-NEC mechanism is simulated as in **Figure 3E** but with a secondary transposase binding site that inhibits transposition when occupied. 0, the secondary site has no affinity for the transposase; 1×, the secondary site has the same affinity for transposase as the primary binding sites at the transposon ends; 2×, secondary site with twice the affinity for transposase. (**B**) The dimerization end occlusion model for OPI. (**C**) All possible states of the DEO model are illustrated. A and B are transposon ends, T is an active transposase monomer, $T_2$ is an inhibitory transposase dimer. Solid lines indicate reaction pathways allowed in the model. The active species is shown in red. (**D**) Simulation of the DEO model using in vitro, non-allosteric parameters as in **Figure 1C**. Inhibitory dimers bind transposon ends with the same affinity as the active monomers. Active monomers dimerize with the same affinity as they bind transposon ends. (**E**) DEO model as in part (**D**) but with in vivo slow diffusion rate as in **Figure 3E**. Affinity of the monomers in the inhibitory dimer is doubled as indicated. (**F**) DEO model as in part (**E**) showing the effect of doubling in the genome size and nuclear volume. (**G**) DEO model as in part (**E**) showing the effect of changing the association rate constant of the active monomers and the inhibitory dimers by the indicated amount.

genetic drift. Later, as the copy number rises, a progressively lower activity per element would prevent excessive damage to the host (e.g. **Townsend and Hartl, 2000**; **Le Rouzic and Capy, 2005**). In the ASO model for mariner transposition, the rate of the reaction increases faster than the copy number at the start of a genomic invasion (**Figure 3A,G,E**). However, later when a number of copies exist in the genome, and the transposase concentration is above a certain threshold, the increased transpositional activity provided by each new copy is almost exactly balanced by the increased competition for binding sites at the transposon ends (**Figure 3E,F**). We note that the competition is conceptually similar to the immunological prozone phenomenon, which has been predicted to apply widely in biological systems that involve bridging interactions (**Bray and Lay, 1997**).

## The cell cycle, and the soma vs the germ line

Transposition in somatic cells leaves no trace in the genomic fossil record and does not help the vertical spread of the transposon through the population. It is also detrimental to the host, particularly in organisms where there is a risk of cancer. Furthermore, since somatic cells may spend a long time in the G1 phase of the cell cycle, when double strand breaks are processed by non-homologous end joining (NHEJ), transposition may cause a net decrease in the number of transposon copies in the event of a failure to reintegrate (**Figure 3H**). The only selective advantage of transposition in soma that we can envisage is that it provides an opportunity for integration into a vector, which may mediate horizontal transfer. Indeed, mariner transposes readily in the soma and is among the most widespread of elements in nature. In contrast, the P element has a limited phylogenetic distribution and its activity is restricted to the germ line (**Engels, 1983**).

Transposition in the germ line aids the vertical spread of the transposon in the population and leaves a genomic fossil record of some of the events, which we still observe today. Cut-and-paste transposition is favored in the germ line because homologous recombination, which is required to reinstate the donor site, is more prevalent than in the soma throughout all phases of the cell cycle (*Robert et al., 2008*; *Tichy et al., 2010*; *Serrano et al., 2011*). Synchronization of transposition with S phase of the cell cycle would provide a further advantage due to the general up-regulation of homologous recombination and the chance of integrating ahead of a replication fork. This strategy was first observed in the bacterial IS*10* (*Roberts et al., 1985*). P element transposition may also be linked to the cell cycle because it preferentially integrates near origins of replication, which will ensure its early replication (*Spradling et al., 2011*). A counter example is that expression of SB transposase is reported to extend G1 and promote the repair of excision sites by NHEJ in the soma (*Walisko et al., 2006*). It is unknown whether mariner transposition is synchronized with the cell cycle. However, this seems plausible because Hsmar1 transposition is greatly stimulated by negative supercoiling (*Claeys Bouuaert et al., 2011*), which probably exists transiently behind an advancing replication fork.

## The rate of synapsis is a dominant factor

The values of the kinetic parameters determine the magnitude of the steady-state rate of transposition as well as the copy number at which it is achieved. The rate of synapsis is a dominant factor, and is in turn influenced by the multimeric state of the transposase, the rate of diffusion and, in the case of Hsmar1, by the allostery between the subunits. At the start of an invasion, a preformed transposase multimer over-comes the improbability of two monomers simultaneously binding opposite ends of the same element when their concentration is low. The advantage arises from the continuity of the DNA between the transposon ends, which ensures that the first dimer-bound-site has a high concentration with respect to its partner. The molar concentration of one DNA site with respect to another is abbreviated as $j_M$, and is about 55 nM for the ends of a 1.3 kb transposon, such as Hsmar1. The magnitude of this value dictates the effectiveness of the ASO mechanism. In dilute solutions, where synapsis is fast, OPI is not significant until the free transposase concentration is a significant fraction of $j_M$. This is evident in *Figure 1K* where 50% inhibition is not achieved until there are $10^7$ copies of the transposon, which corresponds to 40 haploid human-genomes of DNA. However, under the slow diffusion regime that prevails in vivo, OPI becomes significant much earlier, and the steady state is achieved when there are about 200 copies of the transposon (*Figure 3A*). Allostery further increases the effectiveness of ASO by reducing the affinity of the developing transpososome for the second transposon end. This increases the competition for binding sites to such an extent that the steady-state rate is established when there are only a few copies of the transposon (*Figure 3G*).

## Evolutionary considerations

The influence of $j_M$ on the rate of transposition helps to explain the success of miniaturized transposons, which often greatly outnumber their cognate master elements (e.g. *Yang et al., 2009*). However, these relationships have additional complexities. Osmar14 is an autonomous transposon in the rice genome and provides the cognate transposase for Ost35, a more numerous miniature element (*Yang et al., 2009*). This relationship is not obvious from a comparison of the respective transposon end sequences, which appear quite divergent. Our finding that OPI is relieved by reducing the affinity between the transposase and the transposon end provides a further explanation for the unexpected efficiency of this cross-mobilization in addition to their short length (*Figure 4A*).

Another factor that may explain why Osmar14 is less numerous than Ost35 is that it has a secondary transposase binding site near its 3′-end, which inhibits transposition and may represent a mechanism of autoregulation (*Yang et al., 2009*). However, as noted already, a repressive binding site such as this does not change the underlying dynamics of ASO (*Figure 7A*), nor does it dampen exponential amplification in the absence of ASO or a similarly effective mechanism (*Figure 1E*). Another type of secondary-site model is one in which transposase represses its own expression. This is conceptually similar to the cis-action observed in some bacterial systems, in that it prevents the concentration of transposase rising in proportion to the transposon copy number. The weakness of such models is that they only prevent the transposase concentration from rising and can never cause a reduction (*Townsend and Hartl, 2000*). For example, expression of the P element transposase in the soma is repressed in part by binding of a truncated splice variant. However, co-expression of the truncated transposase with the full-length version in the germ line is not sufficient to establish the repressive P-cytotype, which probably requires piwi RNA arising from specific telomeric copies of the element (*Misra and Rio, 1990*; *Brennecke et al., 2008*; *Jensen et al., 2008*).

A more general argument against the effectiveness of 'secondary-site' models is that they are vulnerable to selfish exploitation by any element that lacks the site. Thus, the burden of Ost35 transposition, which lacks the secondary site, would presumably negate any advantage accruing to Osmar14 as a result of its self restraint. ASO is less vulnerable than the secondary-site models because it relies on a competition between active multimers of the recombinase for their primary binding sites at the transposon ends. Although it can, in principle, be subverted by the resistance of low-affinity elements to OPI, this effect can only operate within limits (*Figure 4B*, note the slow amplification when the copy number is low). Although not completely immune, the ASO mechanism is robust to this type of selfish exploitation. ASO also provides a homeostatic mechanism in that a doubling of the transposase concentration always halves the rate of transposition per element, irrespective of the values of the kinetic constants governing the reaction. A larger genome will thus always receive a proportionately higher rate of transposition. In the case of mariner, the relative genetic burden is thus independent of the host genome size and is dictated by adaptive features of the transposon itself. These features include the transposase binding affinity, the strength of the promoter in a given host and the strength of the allosteric coupling between subunits. The latter is a powerful determinant able to change the rate of synapsis over many orders of magnitude. Indeed, allostery provides significant adaptive flexibility, which is unavailable in the S-PD reaction mechanism. Finally, by buffering the rate of transposition against changes in the transposase concentration, ASO may also provide a degree of protection from post-transcriptional silencing and heterochromatinization responses.

## Materials and methods

### Protein expression, purification and in vitro assays

The wild type and RA104 transposases were expressed from pRC880 and pRC1128, respectively (*Claeys Bouuaert and Chalmers, 2010*). Unless stated otherwise, each 50 µl transposition reaction contained 6.7 nM of plasmid substrate and 20 nM transposase in 20 mM Tris-HCl pH8, 100 mM NaCl, 2 mM DTT, 2.5 mM $MgCl_2$ and 10% glycerol. The standard substrate was pRC650, which encodes a mini-transposon with 30 bp Hsmar1 transposon ends. Plasmid pRC919 has a single 30 bp Hsmar1 end cloned into the pBluescript polylinker. Open circular substrates were prepared using the Nb.BsrDI endonuclease, which nicks the plasmids at several sites some distance away from the transposon ends. Transposition reactions were incubated at 37 °C and analyzed by loading 400 ng of the DNA on each lane of a TBE-buffered 1.1% agarose gel. After electrophoresis, the gel was stained with ethidium bromide, destained in water, and photographed on a 310 nm transilluminator using a DC290 camera (Kodak, Rochester, NY) with a 590 DF bandpass filter. Digital photographs were quantified using the Image Gauge software package ScienceLab 2003 (Fuji Corporation, Tokyo, Japan).

In the EMSA 97 bp DNA fragments carrying Hsmar1 transposon ends were prepared by digesting pRC919 with XmaI and labeled at both ends using [α-$^{32}$P]dCTP and the Klenow enzyme. Unless stated otherwise each 20 µl reaction contained 250 ng of non-specific plasmid DNA as a carrier, 2 nM labeled substrate and 6 nM transposase. Complexes were assembled for the indicated times in a buffer containing 20 mM HEPES pH7.5, 100 mM NaCl, 2 mM DTT, 10% glycerol, 250 µg/ml BSA. Products were separated on 5 or 7% polyacrylamide Tris-acetate-EDTA gels.

### In vivo assays

Transposition assays in HeLa cells were performed as described by (*Miskey et al., 2007*; *Grabundzija et al., 2010*) using isogenic plasmids, except that the SB transposase gene and transposon ends were replaced by the respective sequences from PB and Hsmar1.

### Development and implementation of the computer model

The kinetic models for interactions between transposase and DNA is presented in *Figure 1B,I*. A and B represent the identical inverted repeat sequences flanking the transposon, which are the specific binding sites of the transposase. The hyphens joining A and B represent the transposon DNA between the inverted repeats. $T_2$ represents transposase, which is always dimeric in this model, $T_{2,nsb}$ represents non-specifically bound transposase, $T_{2,free}$ represents freely diffusing transposase, AT and BT represent transposase monomers bound to A or B, respectively, and $AT_2$ and $BT_2$ represent transposase dimers bound to A or B, respectively. The flux through the reaction was simulated in the Macintosh version of MATLAB R2010b using the ordinary differential equation solver ode15s.

For the mariner model, we begin by considering an idealized situation in which we make two significant assumptions. Firstly, that since the A and B sites are identical, they have identical binding kinetics. Secondly, that the two DNA-binding sites on a single transposase dimer have identical behavior and do not interact with each other, meaning that there are no allosteric interactions between the transposase subunits. In this idealized situation the first and second transposon end to be bound by the transposase, which ever this may be, are bound with equal affinity and the rate depends only on the relative concentration of the reactants. Later we will also consider a model in which allosteric interactions between the transposase subunits causes a kinetic barrier to recruitment of the second transposon end.

Because of these assumptions mentioned above, our model only needs to include the reaction rate constants shown in *Figure 1I*, which are $k_0$, $k_{-0}$, $k_1$, $k_{-1}$, $k_2$, $k_{-2}$, and $k_3$. Of these, $k_0$ and $k_1$ are second order rate constants (units of $M^{-1}s^{-1}$) and the rest are first order rate constants (units of $s^{-1}$). The experimental literature provides reasonable estimates for $k_{-0}/k_0$, $k_1$ and $k_{-1}$, given below, but not for $k_2$, which we go on to calculate. $k_3$ is also relatively unknown, which we discuss later.

For the synapsis-by-protein-dimerization model, as exemplified by Tn10 and Tn5, we make the same assumptions as detailed above, except that the free transposase is monomeric.

In our model, protein expression is instantaneous, producing a specified number of transposase dimers per transposon copy. Unless stated otherwise this is 500 dimers in the mariner model and 1000 monomers in the prokaryotic model. Instantaneous expression is necessary to simulate the reaction under the fast diffusion regime which prevails in vitro. Under the in vivo slow-diffusion regime, the reaction is so slow that the dynamics of protein expression and degradation are irrelevant.

## Choice of the kinetic parameters for DNA binding–*k*0, *k*−0, *k*1, and *k*−1

The Hsmar1 and Mos1 transposases are 37% identical and align throughout their entire length with only a single residue indel. The respective proteins will therefore have very similar three dimensional structures. A crystal structure for the Mos1 transposase revealed that it has an N-terminal DNA binding domain with two helix-turn-helix (H-T-H) motifs that are inserted into the major groove of the DNA close to the end of the transposon (*Richardson et al., 2009*). We can therefore expect that the dissociation constant for transposon end binding ($K_d = k_{-1}/k_1$) will be in the mid-picomolar concentration range. The lactose repressor (LacI) is one of the most thoroughly characterized H-T-H proteins, with many studies addressing the kinetic parameters of DNA binding. The LacI dimer is similar to a monomer of Hsmar1 transposase in that it binds to operator sequences by a pair of H-T-H motifs, which are located in adjacent major grooves. We therefore elected to use the kinetic parameters for LacI to model transposase binding to the transposon end. Estimated parameters for the behavior of the transposase are perhaps not ideal. However, our conclusions do not depend on the absolute values of the parameters because they have minimal effect beyond changing the scale on the axes of the graphs generated by the simulation. The relationships between the various parameters, which provide the key insights of the work, remain unchanged, as explained further below and in the main text.

The values for the lactose repressor's dissociation constant quoted in the textbooks and the literature range from about $10^{-9}$ to $10^{-14}$ M. However, it is now generally accepted that the $K_d$ for LacI, and other dimeric H-T-H proteins, binding to their primary sites is in the mid to high picomolar concentration range. Many of the very low values quoted in the literature date from the late 1970s and early 1980s, and were determined at unphysiologically low salt concentrations. Another factor that complicates the lactose repressor literature is the complexity of the multiple binding sites encoded by the operator, which promote DNA looping and the mutual stabilization of dimers bound in cis to each other. Here we have used the kinetic parameters determined by Wells and Matthews and colleagues for LacI binding to a single site encoded on a 40 bp fragment of the lac operator under a physiological salt concentration (Table 3 of *Hsieh et al. (1987)*).

In the course of our experiments with Hsmar1 transposase we have never recorded significant inhibition of the reaction by the moderate amounts of non-specific DNA used as a target in many of our experiments. This suggests that the affinity of the transposase for non-specific DNA is relatively low. We have therefore elected to adopt the equilibrium constant ($K_d = k_{-0}/k_0$) of LacI for non-specific DNA because it is probably fairly typical for trans-acting DNA binding proteins. We adopted the value provided by Von Hippel and colleagues because it was determined at a similar physiological salt concentration to the parameters noted above for site-specific binding (*Revzin and Von Hippel, 1977*). We used this equilibrium constant to assign arbitrary association and dissociation rate constants,

**Table 1.** Kinetic parameters for DNA binding

| Reaction | $K_d$ (M) | $k_0$, $k_1$ ($M^{-1}s^{-1}$) | $k_0$, $k_1$ ($\mu m^3\ s^{-1}$) | $k_{-0}$, $k_{-1}$ ($s^{-1}$) | $t_{1/2}$ (s) |
|---|---|---|---|---|---|
| LacI specific binding | $3.1 \times 10^{-11}$ | $3.8 \times 10^8$ * | 0.633 | $1.2 \times 10^{-2}$ * | 58 † |
| LacI non-specific binding | $1.4 \times 10^{-5}$ ‡ | $9.9 \times 10^6$ § | 0.0166 | 139 § | $5 \times 10^{-3}$ † |

*From Table 3 of **Hsieh et al. (1987)**.

†This is the experimental estimate for LacI bound to non-specific DNA dissociating to the bulk solution (**Elf et al., 2007**). During this time it will have visited about 85 non-specific sites by one-dimensional diffusion. However, for the purposes of the simulation it is convenient to subsume the two phases into a single bulk behavior of the system (see text for details).

‡From **Revzin and Von Hippel (1977)**.

§Estimated from $K_d$ using the relationship $K_d = k_{-1}/k_1$ or $K_d = k_{-0}/k_0$.

---

$k_0$ and $k_{-0}$. It is difficult to define these rate constants precisely because DNA binding proteins have two distinct modes of non-specific interaction. Firstly, binding from the bulk solution is probably rapid with a rate equal to a sizable fraction of a diffusion-limited reaction. This is followed by a one-dimensional diffusion phase, during which the protein visits non-specific sites that are considered to be 1 bp apart. The equilibrium constant, $K_d$, is thus the product of one- and three-dimensional events, each of which has its own $k_{on}$ and $k_{off}$ components. Fortunately, the absolute values of $k_{-0}$ and $k_0$ are not important in the simulation because they serve only to define the equilibrium constant for non-specific sites, which in turn specifies the concentration of free transposase available for transposon end binding in a genome of a given size. The validity of this assumption was confirmed by running the simulation using several different pairs of values for $k_0$ and $k_{-0}$: the output was unchanged provided that $k_{-0}/k_0$ was always equal to the experimentally determined $K_d$ (not shown).

The values for the specific and non-specific parameters are tabulated in **Table 1**, where they are also converted into the absolute units used in the simulation (i.e., numbers of molecules per cubic micrometer). The dissociation constant for non-specific DNA provided by Von Hippel and colleagues is expressed in units of per base pair. We have assumed a haploid genome size of $3 \times 10^9$ bp, which provides $6 \times 10^9$ non-specific binding sites per diploid genome. The volume of the nucleus is assumed to be 500 $\mu m^3$.

## Rate of synapsis, *k2*

We calculated the first-order reaction rate constant $k_2$ as follows. Throughout this discussion, $T_2$ represents freely diffusing transposase, which is shown as $T_{2,free}$ in the model diagram.

Consider the two elementary reactions from our model that are shown at the bottom-left and the top-right regions of **Figure 1I**, respectively. Ignoring the transposon's internal DNA sequences for now, these reactions are (1) $T_2 + B \rightarrow BT_2$ and (2) $AT_2 + B \rightarrow AT_2B$. Both reactions involve the binding of transposase to DNA site B, so they are chemically identical. This might suggest that they would have the same reaction rates. However, in actuality, their reaction rates differ because the rate of diffusion for the free transposase is different from that of the DNA-bound transposase. Expressing their reaction rate constants as $k_1$ for reaction 1 (as in the model diagram) and $k_1'$ for reaction 2, the respective reaction rates are

$$\frac{d[BT_2]}{dt} = k_1[T_2][B],\tag{1}$$

$$\frac{d[AT_2B]}{dt} = k_1'[AT_2][B].\tag{2}$$

The Collins and Kimball reaction rate theory (**Rice, 1985**) enables us to expand these reaction rate constants into one contribution that arises from diffusion and a second that arises from the binding activation energy. According to this theory, the expanded reaction rate constants are

$$\frac{1}{k_1} = \frac{1}{k_{1,diff}} + \frac{1}{k_{act}},\tag{3}$$

$$\frac{1}{k_1^{'}} = \frac{1}{k_{1,diff}^{'}} + \frac{1}{k_{act}}, \tag{4}$$

where $k_{1,diff}$ and $k_{1,diff}^{'}$ are diffusion-limited reaction rate constants and $k_{act}$ is an activation-limited reaction rate constant. These equations include the same $k_{act}$ value because we assumed above that the two reactions are chemically identical; under this assumption, they should have the same binding activation energy and thus the same activation-limited reaction rate constants.

The necessary diffusion-limited reaction rate constants can be calculated from equations that Berg derived for the diffusion-limited association rates of proteins to DNA (**Berg, 1984**). Berg's equation 16 is for the association rate of a freely diffusing protein to a specific DNA site, while his equation 27 is for the association rate of a DNA-bound protein to another specific DNA site. In our notation, these equations are

$$k_{1,diff} = 4\pi D_p R + D_s a \left(\frac{R}{a}\right)^{1/3}, \tag{5}$$

$$k_{1,diff}^{'} = 1.4 D_s a \left(\frac{R}{a}\right)^{1/3}. \tag{6}$$

These equations were derived by treating DNA with the worm-like chain model, in which DNA is represented as a long thin semi-flexible filament that bends smoothly over the course of its length. This model was recently shown to be reasonably accurate for modeling DNA dynamics (**Petrov et al., 2006**). The central model parameter is the filament persistence length, $a$, which is the characteristic length for filament bending. According to the model, DNA binding sites diffuse rapidly within their local regions, while also gradually diffusing away to more distant regions. The rate of this local 'segmental diffusion' is quantified with the diffusion coefficient $D_s$. $D_s$ is the translational diffusion coefficient of a free DNA fragment with length $a$ (**Berg, 1984**). Continuing with the above equations, $R$ represents the distance over which a specific interaction occurs between transposase and its DNA binding sites. It arises from the assumption that the two reactants (the transposase binding domain and a DNA binding site, in this case) can be treated as hard spheres that react upon collision. Finally, $D_p$ is the diffusion coefficient of free transposase relative to the center of mass of the DNA.

Two final equations need to be introduced to enable us to calculate the rate of synapsis, $k_2$. The reaction rate equation for reaction 2, given in **equation 2**, can be rearranged by grouping $k_1^{'}$ and [B],

$$\frac{d[AT_2B]}{dt} = (k_1^{'}[B])[AT_2] = k_2[AT_2]. \tag{7}$$

The latter equality defines $k_2$, a pseudo-first order reaction rate constant, as $k_1^{'}[B]$. Here, [B] is the concentration of the B DNA binding site in the vicinity of the A binding site. We used the following empirical equation from **Ringrose et al. (1999)** to find [B]:

$$j_M = \left(\frac{4a}{10^4 b}\right)^{3/2} \exp\left(\frac{-460a^2}{6.25b^2}\right)\left(\frac{1.25 \cdot 10^5}{a^3}\right). \tag{8}$$

$j_M$ is the local concentration of one DNA site in the vicinity of another in M, $a$ is the DNA persistence length in nanometers, and $b$ is the separation between the sites in base pairs.

**Table 2** lists the values that we used and derived to estimate $k_2$. In brief, we used **equation 3**, including an experimental value for $k_1$ and a calculated value for $k_{1,diff}$ from **equation 5**, to calculate $k_{act}$. We then substituted $k_{act}$ into **equation 4**, along with a calculated value for $k_{1,diff}^{'}$ from **equation 6**, to estimate $k_1^{'}$. Finally, we used **equations 7** and **8** to estimate $k_2$, the rate of synapsis, from $k_1^{'}$.

These theoretical considerations allow us to develop a model of an idealized transposition reaction, in which the monomers within a transposase dimer bind the first and second transposon ends with equal affinity. In this idealized situation, synapsis of the transposon ends is simply a product of the

**Table 2.** Estimation of $k_2$ for transposase-DNA interactions in vitro

| Quantity | Symbol | Value | Source |
|---|---|---|---|
| Reaction 1 rate constant | $k_1$ | $3.8 \times 10^8$ M$^{-1}$s$^{-1}$ | Table 3 of **Hsieh et al. (1987)**. |
| DNA persistence length | $a$ | 50 nm | Well established (e.g. **Petrov et al., 2006**). |
| Reaction radius | $R$ | 2.5 nm | The transposase dimeric binding domain is 111 amino acids and weighs 26 kDa. From Note 3 of **Andrews (2012)**, its radius is about 1.9 nm. Add to this about 0.6 nm to account for the 0.34 nm length of 1 DNA basepair (because transposase specificity is accurate to 1 bp), combined with the 1 nm radius of DNA. |
| Segmental diffusion coefficient | $D_s$ | 27 µm$^2$s$^{-1}$ | Figure 3 of **Petrov et al. (2006)** for 147 bp, which is 1 DNA persistence length. |
| Transposase diffusion coefficient | $D_p$ | 61 µm$^2$s$^{-1}$ | Note 3 of **Andrews (2012)**, using a molecular weight of 80.7 kDa for transposase. Assumes DNA center of mass is effectively stationary. |
| Diffusion-limited part of $k_1$ | $k_{1,diff}$ | $1.5 \times 10^9$ M$^{-1}$s$^{-1}$ | **Equation 5**. |
| Activation-limited reaction rate | $k_{act}$ | $5.2 \times 10^8$ M$^{-1}$s$^{-1}$ | **Equation 3**. |
| Diffusion-limited part of $k_1'$ | $k_{1,diff}'$ | $4.2 \times 10^8$ M$^{-1}$s$^{-1}$ | **Equation 6**. |
| Reaction 2 rate constant | $k_1'$ | $2.3 \times 10^8$ M$^{-1}$s$^{-1}$ | **Equation 4**. |
| Effective local DNA site concentration | [B] | $5.5 \times 10^{-8}$ M | **Equation 8**, using a 1287 bp transposon length. |
| Transposase-transposon end dissociation rate constant | $k_{-1}$ | $5.8 \times 10^{-4}$ s$^{-1}$ | Estimated from **Figure 3C**. |
| Reaction 2, pseudo-first order rate constant for synapsis | $k_2$ | 12.7 s$^{-1}$ | **Equation 7**. |
| Reaction 2, pseudo-first order rate constant for synapsis with an open circular substrate in vitro. | $k_2$ | $9.6 \times 10^{-5}$ s$^{-1}$ | Estimated using the equation $t_{1/2} = \ln2/k_2$. $t_{1/2}$ is the time taken to consume one quarter of the substrate in **Figure 3D**. See **Figure 2—figure supplement 1** for an explanation of why the time to consume one quarter of the substrate is used, rather than the time to consume half of the substrate. |

sequential binding of the transposase dimer to sites at opposite ends of the element. We then go on to consider a more realistic model in which allosteric interactions between the subunits reduces the affinity of the developing transpososome for the second transposon end. The magnitude of this effect is provided by experimental estimates of the rate of synapsis (**Figure 3D**, **Table 2**; **Claeys Bouuaert and Chalmers, 2010**; **Claeys Bouuaert et al., 2011**).

In **Table 2**, the reaction radius, $R$, is a fairly rough estimate. However, this is not a major concern because reaction rates are relatively insensitive to the reaction radius (**Berg, 1984**) (they scale as $R^{1/3}$). Other values are likely to be more accurate. A result of the calculations shown there is that $k_1'$ is only about a factor of 1.6 slower than $k_1$. This indicates that the slower diffusion of DNA-bound transposase than of free transposase has only modest impact for in vitro experiments.

For Tn10/5 there is no experimental data regarding the affinity of the transposon-end-bound monomers when they collide with each other and achieve synapsis (see kinetic model in **Figure 1B**). For the purposes of the simulation we assigned an association rate constant equal to that for the mariner transposase binding to a transposon end. In the respective prokaryotic and eukaryotic models, the rate of synapsis is therefore determined by the same association rate constant and the rate of diffusion, which is identical in both systems. The outputs of the respective simulations are therefore directly comparable and reveal differences in the underlying kinetic models.

**Table 3.** Estimation of in vivo transposase-DNA interactions

| Quantity | Symbol | Value | Source |
|---|---|---|---|
| DNA persistence length | $a$ | 50 nm | See *Table 2*. |
| Reaction radius | $R$ | 2.5 nm | See *Table 2*. |
| Segmental diffusion coefficient | $D_s$ | $5 \times 10^{-4}$ µm$^2$s$^{-1}$ | From Figure 2 caption of *Marshall et al. (1997)*. |
| Transposase diffusion coefficient | $D_p$ | 15 µm$^2$s$^{-1}$ | ¼ of *Table 2* value, based on Note 3 of *Andrews (2012)*. |
| Diffusion-limited part of $k_1$ | $k_{1,diff}$ | $2.9 \times 10^{8}$ M$^{-1}$s$^{-1}$ | *Equation 5*. |
| Activation-limited reaction rate | $k_{act}$ | $5.2 \times 10^{8}$ M$^{-1}$s$^{-1}$ | From *Table 2*. |
| Diffusion-limited part of $k_1'$ | $k_{1,diff}'$ | $7.8 \times 10^{3}$ M$^{-1}$s$^{-1}$ | *Equation 6*. |
| Reaction 1 rate constant | $k_1$ | $1.9 \times 10^{8}$ M$^{-1}$s$^{-1}$ | *Equation 3*. |
| Reaction 2 rate constant | $k_1'$ | $7.8 \times 10^{3}$ M$^{-1}$s$^{-1}$ | *Equation 4*. |
| Effective local DNA site concentration | [B] | $5.5 \times 10^{-8}$ M | See *Table 2*. |
| Reaction 2 pseudo-first order rate constant | $k_2$ | $4.3 \times 10^{-4}$ s$^{-1}$ | *Equation 7*. |

## The effects of macromolecular crowding in vivo

Thus far we have considered reactions in dilute solution. To provide a more realistic description of the reaction in vivo we next had to account for macromolecular crowding, which reduces the rate of diffusion. We therefore computed the same numbers that are shown in *Table 2*, but for the in vivo situation, which are shown in *Table 3*. The only parameters sensitive to the high viscosity in vivo are the association rate constants, which includes the rate of synapsis. Our calculations show that the association rate constant, $k_1$, suffers a modest twofold penalty in vivo (*Table 3*). This relieves OPI and increases the rate of transposition twofold (not shown). However, this is precisely offset by the penalty on non-specific binding, $k_0$, which achieves the opposite. The results are very different for the rate of synapsis, which suffers an in vivo diffusion penalty of $2.8 \times 10^{4}$-fold (*Tables 2 and 3*). However, many DNA interactions are known to occur over much longer-ranges than those between transposon ends. It is therefore possible that other factors in vivo mitigate the severity of the diffusion penalty. For example, there is evidence that the factor $j_M$ is doubled by the chromatinization of DNA (*Ringrose et al., 1999*). The in vivo values in *Table 3* should therefore be seen as initial estimates rather than an exhaustive treatment of the topic. Nevertheless, our calculated rate of synapsis is very close to the rates of lox/cre recombination measured in vivo with similarly separated recombination sites (*Ringrose et al., 1999*).

## Synapsis unbinding, *k–2*

In the simplest model for synapsis the rate of the reverse reaction, $k_{-2}$, would be equal to the rate of transposon end unbinding, $k_{-1}$ (see *Figure 1I*). However, in practice we consider synapsis to be essentially irreversible for the following reasons. Transposition reaction mixtures in which the catalytic Mg$^{2+}$ ions are replaced by Ca$^{2+}$ support transpososome assembly, but none of the chemical steps of the reaction. After the addition of the catalytic metal ion, the first nick at the transposon end is detected immediately (*Claeys Bouuaert and Chalmers, 2010*). This shows that nicking is very much faster than synapsis or synapsis unbinding. Synapsis in the presence of the catalytic metal ion is therefore essentially irreversible because of the loss of enthalpy associated with hydrolysis of the phosphodiester bond. In principle, it is possible that there exists some form of unstable synaptic complex that matures into the stable form that immediately precedes the first chemical reaction. If such a complex existed it would be irrelevant to our experiments, in which the rate of synapsis is estimated from the rate of the first nick. When we consider the rate of synapsis we are therefore restricting ourselves to 'productive synapses', which yield the first nick. For the purposes of the simulation we have set the rate constant for synapsis unbinding ($k_{-2}$) at $10^{-10}$ s$^{-1}$. Synapsis in the model is therefore essentially irreversible.

## Allostery and experimental estimates of $k2$ and $k-1$

When transposase dimer binds to the first transposon end it undergoes a conformational change that lowers the affinity of the unoccupied DNA binding domain for the second transposon end compared to the first (for details see main text and *Claeys Bouuaert et al., 2011*). This means that the actual value of $k_2$ will be much lower than that calculated above for the idealized reaction, in which synapsis is the product of sequential, chemically identical, collisions between DNA binding domains and transposon ends. To account for the allosteric interactions between the transposase subunits we therefore require experimental estimates of $k_2$ and $k_{-1}$.

Estimates for $k_{-1}$ and $k_2$ are provided by the kinetics of the in vitro reaction. In *Figure 3C* we see that about 50% of the transposon end present in SEC2 is released after 20 min incubation in the presence of cold competitor DNA. This corresponds to a dissociation rate constant of $5.8 \times 10^{-4}$ s$^{-1}$. We have previously shown that first strand nicking depends on, and is much faster than, synapsis (*Claeys Bouuaert et al., 2011* and 'Discussion' above). Consumption of the supercoiled substrate therefore provides an estimate of the rate of synapsis. Under optimal reaction conditions, with about one dimer of active transposase per transposon, only one half of the substrate is initially occupied by a single dimer and is therefore able to react. As explained above, the random association of transposase dimers and transposon ends at the start of the reaction means that a quarter would be occupied by two dimers and would suffer OPI and the other quarter would be completely unoccupied. The time required to consume one quarter of the substrate therefore approximates the half-time for synapsis. Since eukaryotic DNA has very little free supercoiling, the most relevant rate of synapsis is perhaps provided by an open circular substrate. In *Figure 3D*, the time taken to consume one quarter of the open circular substrate is about 2 hr, which corresponds to a pseudo-first order rate constant of $9.6 \times 10^{-5}$ s$^{-1}$. Note that the reason we are able to estimate the rate of synapsis is owing to the optimal reaction condition in vitro where $k_2$ is largely independent of $k_0$, $k_{-0}$, $k_1$ and $k_{-1}$. This is because there is very little non-specific DNA present and the transposase concentration is such that most dimers are engaged in productive interactions with transposons.

## Rate of cleavage, integration and maturation of the product, $k_3$

The rate limiting steps of the reaction with various substrates has been determined previously (*Claeys Bouuaert and Chalmers, 2010*; *Claeys Bouuaert et al., 2011*). With relaxed plasmid and short linear substrates, synapsis is the rate limiting step. Supercoiling in the substrate accelerates synapsis because the transposon ends have a high relative concentration in the plectosome and a favorable angular distribution. In an in vitro reaction, where the plasmid has more than twice its natural level of free supercoiling, the rate of synapsis is faster than cleavage of the second strand at the transposon end. Note that second strand cleavage at the second transposon end releases the transposon from the donor site (illustrated in *Figure 2A*). In a kinetic analysis of staged in vitro reactions with a supercoiled substrate, a small amount of excised transposon is observed at early time points [this is below the region of the gels shown in *Figure 2B*, but can be seen in Figure 5B of *Claeys Bouuaert et al. (2011)*]. The rate of integration must therefore be faster than the rate of second strand cleavage. The half life of the excised transposon is approximated by the time required to convert a quarter of the supercoiled substrate into product. From *Figure 2B* and other experiments this appears to be about 8 min which corresponds to a rate constant for integration of $1.4 \times 10^{-3}$ s$^{-1}$.

Although this value for $k_3$ is appropriate in the simulations using in vitro parameters, it would be unrealistically short in an in vivo situation where the integration complex must be disassembled and the empty donor site restored. However, our reference in vitro simulation (*Figure 3E*) is relatively insensitive to the value of $k_3$. This is because the extent to which the Hsmar1 specific factors, namely the slow dissociation rate of SEC2 and the slow recruitment of the second end, slow synapsis. Furthermore, once a steady-state rate is established the length of the maturation process does not matter because the rates of pathway entry and exit are equal. Thus, if $k_3$ is reduced by three orders of magnitude to $1.4 \times 10^{-6}$ s$^{-1}$ ($t_{1/2}$ = 5.7 days), the final rate of transposon amplification does not change significantly. However, it does take slightly longer to be achieved because of the time required for pathway entry and exit rates to balance (not shown). Note that we assume that every transposon that initiates catalysis goes on to complete excision and integration. This is supported by the in vitro reaction systems for Hsmar1 and Tn10 (*Chalmers and Kleckner, 1994*, *1996*; *Claeys Bouuaert and Chalmers, 2010*; *Claeys Bouuaert et al., 2011*). The efficiency of the reaction in vivo is unlikely to be so high and this is dealt with in *Figure 3G* and the section of text entitled 'The efficiency of transposition in vivo.'

## Transposase expression level

Many transposons, and mariner in particular, have a wide host range. The promoters driving expression of transposase genes are therefore assumed to rely on general transcription factors. However, there is little published data pertaining to the possible strength of the promoters. Expression of the Hsmar1 promoter has been detected using a luciferase reporter system, but quantification is lacking for the contribution of a single copy of the element (*Miskey et al., 2007*). A recent proteomic study quantified the abundance of 5000 vertebrate proteins in mouse cells (*Schwanhausser et al., 2011*). The copy number per cell for most proteins ranged from about 100 to 1 million: The median was 16,000 and the mode was about 5000. In the simulation we set the transposase expression at 500 dimers per transposon. Although this is at the low end of the range for vertebrate proteins, it is not unusually low and allows for up to 1000 copies of the transposon before protein levels would reach the high end of the range. However, we note, once again, that our conclusions do not depend on the absolute values of the parameters because they have minimal effect beyond changing the scale on axes of the graphs generated by the simulation. The relationships between the various parameters, which provide the key insights of the work, remain unchanged.

## Acknowledgements

We would like to thank Yves Bigot, Nancy Kleckner, Boyan Bonev, Steve Halford, Steve Kowalczykowski, Csaba Miskey, Zoltan Ivics, Deepesh Agarwal, and Andrew Howlett for helpful advice; Neil Walker for technical assistance; and Mark Calleja and Jenny Barna for maintaining CamGrid. We would also like to thank the referees for valuable contributions to this work. This publication is dedicated to the memory of Peter and Barbara Blakey whose generous bequest constructed the laboratory infrastructure.

## Additional information

### Funding

| Funder | Grant reference number | Author |
|---|---|---|
| Wellcome Trust | 093160/Z/10/Z | Corentin Claeys Bouuaert, Danxu Liu, Ronald Chalmers |
| Royal Society | | Karen Lipkow |
| National Institute of General Medical Sciences | 91687 | Steven S Andrews |
| National Institutes of Health | R01GM086615-01 | Steven S Andrews |
| Microsoft Research Faculty Fellowship | | Karen Lipkow |
| Apple Research & Technology Support | | Karen Lipkow |
| EU-IndiaGrid2 | | Karen Lipkow |

The funders had no role in study design, data collection and interpretation, or the decision to submit the work for publication.

### Author contributions

CCB, DL, RC, Conception and design, Acquisition of data, Analysis and interpretation of data, Drafting or revising the article, Contributed unpublished essential data or reagents; KL, SSA, Conception and design, Acquisition of data, Analysis and interpretation of data, Drafting or revising the article

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
