## [Decision Letter]

Thank you for sending your work entitled “The autoregulation of DNA transposons in higher eukaryotes: insights from the human mariner element Hsmar1” for consideration at *eLife*. Your article has been favorably evaluated by a Senior editor and 2 reviewers, one of whom is a member of our Board of Reviewing Editors.

The Reviewing editor and the other reviewer discussed their comments before we reached this decision, and the Reviewing editor has assembled the following comments to help you prepare a revised submission.

The fundamental problem at hand is how a transposable element manages to co-exist with its host without its intrinsic tendency for proliferation eventually rendering the host (and thus also the transposon) extinct. As the authors point out, a solution to this problem is known for certain prokaryotic transposons, but the elucidated mechanism does not apply to eukaryotes (because it requires cis-acting transposase). In the case of eukaryotic transposons, there have been only sporadic proposed models, which have not been analyzed or assessed either experimentally or by quantitative modeling (which, in every case is dependent on information regarding the basic biochemistry of the transposition reaction).

The present paper suggests an entirely new model for regulation of eukaryotic transposons, motivated by biochemical findings, with direct implications for the model from theoretical simulations in combination with both in vitro and in vivo analysis of transposition. Numerous extensions of the model, including features relating to the presence/accumulation/roles of non-autonomous elements, are also considered.

The proposed model is very interesting and very sensible, not only a priori in a qualitative way but also quantitatively, as illustrated by modeling. Furthermore, taken together, the presented observations provide very strong support for the validity of the model. The experimental observations are important and compelling, particularly the in vitro findings relating the responses to variations in transposase and transposon end concentrations, and the in vivo verification of several key predictions. No such study has ever been performed for any transposon, even in the prokaryotic cases, and the results presented here should be of broad interest to both the evolutionary biology community and the transposition mechanism community.

Major comments:

1) Figure 1 and related text: We think the reader might appreciate an explanation of why the kinetic schemes shown in Figure 1 result in duplication of the transposon. If we understand correctly, the idea is that the excision of the transposon occurs after the transposon has replicated. The site from which it was excised is repaired by homologous recombination using the other chromosome as a template, regenerating the transposon at the excision site, and the excised transposon goes on to integrate somewhere else. Depending upon whether it integrates before or after the replication fork, or fails to integrate at all, the number of transposons in the genome can increase by different amounts or decrease.

In any case, it would certainly be helpful if the authors were to explain their thinking here, and to clarify the relationships between the kinetic schemes in Figure 1 and the discussion in Figure 3.

2) Figure 7 and related text: we stumbled over this section. The writing makes it seem as though a previously-proposed dimerization end-occlusion model also accounts for stable transposon levels.

Assuming we have understood correctly, we think the situation is as follows: the first paragraph discusses a case with a secondary binding site. It is likely that this site was proposed to have the effect of repressing transposition. The important point, which is implicit but should be stated explicitly, is that this feature does not in any way prevent the accumulation over time, which is the problem at hand here – it simply slows things down, regardless of whether that is in the current mechanism or the S-PD mechanism. That is, you still need the current mechanism (and the S-PD mechanism doesn’t work) if this feature is present.

The second paragraph addresses the DEO model for Tn*5* and mariner. Again, the writing does not make clear what the authors wish to say. The case of Tn*5* is not relevant here because it involves a cis-acting transposase. But the case of mariner obviously is relevant. Here there are two points. First, the DEO model is fine, but the biochemistry presented in this paper for mariner suggests that it is not what is going on. This point should be made explicitly. Second, in principle, an element with a *different* biochemistry might use this mechanism; but, if so, the modeling suggests that there is a longer lag time before the effect kicks in. The authors are perhaps trying to be sure to be generous to the previous considerations. But they need to say (a) DEO can’t be true for mariner because the biochemistry says it isn’t and (b) it could potentially be true for some other element but with a disadvantage. Are there further candidate elements that might work in such a way?

3) This general class of mechanism is formally analogous to the “prozone effect”, which is the reason why complexes formed between a bivalent antibody and a bivalent antigen tend to be dimeric when either antibody or antigen is in excess, and highly polymeric when they are balanced. This mechanism pops up again and again in protein regulation. See, for example Bray E, Lay S, PNAS 1997.

---

## [Author Response]

*1) Figure 1 and related text: We think the reader might appreciate an explanation of why the kinetic schemes shown in Figure 1 result in duplication of the transposon. If we understand correctly, the idea is that the excision of the transposon occurs after the transposon has replicated. The site from which it was excised is repaired by homologous recombination using the other chromosome as a template, regenerating the transposon at the excision site, and the excised transposon goes on to integrate somewhere else. Depending upon whether it integrates before or after the replication fork, or fails to integrate at all, the number of transposons in the genome can increase by different amounts or decrease*.

*In any case, it would certainly be helpful if the authors were to explain their thinking here, and to clarify the relationships between the kinetic schemes in Figure 1 and the discussion in Figure 3*.

Agreed: we have now restructured the first two paragraphs of the Results section. The issues of replication and repair are now covered at the end of the first paragraph.

*2) Figure 7 and related text: we stumbled over this section. The writing makes it seem as though a previously-proposed dimerization end-occlusion model also accounts for stable transposon levels*.

*Assuming we have understood correctly, we think the situation is as follows: the first paragraph discusses a case with a secondary binding site. It is likely that this site was proposed to have the effect of repressing transposition. The important point, which is implicit but should be stated explicitly, is that this feature does not in any way prevent the accumulation over time, which is the problem at hand here – it simply slows things down, regardless of whether that is in the current mechanism or the S-PD mechanism. That is, you still need the current mechanism (and the S-PD mechanism doesn’t work) if this feature is present*.

Agreed: the referees have understood correctly. We have substantially rewritten the first paragraph of this section as two new paragraphs, which make the suggested changes.

*The second paragraph addresses the DEO model for Tn*5 *and mariner. Again, the writing does not make clear what the authors wish to say. The case of Tn*5 *is not relevant here because it involves a cis-acting transposase. But the case of mariner obviously is relevant. Here there are two points. First, the DEO model is fine, but the biochemistry presented in this paper for mariner suggests that it is not what is going on. This point should be made explicitly. Second, in principle, an element with a* different *biochemistry might use this mechanism; but, if so, the modeling suggests that there is a longer lag time before the effect kicks in. The authors are perhaps trying to be sure to be generous to the previous considerations. But they need to say (a) DEO can’t be true for mariner because the biochemistry says it isn’t and (b) it could potentially be true for some other element but with a disadvantage. Are there further candidate elements that might work in such a way?*

Yes, thank you. We always like to try to be as generous as possible. However, we have now rewritten and restructured the last two paragraphs of the results to make the explicit statements suggested by the referees.

*3) This general class of mechanism is formally analogous to the “prozone effect”, which is the reason why complexes formed between a bivalent antibody and a bivalent antigen tend to be dimeric when either antibody or antigen is in excess, and highly polymeric when they are balanced. This mechanism pops up again and again in protein regulation. See, for example Bray E, Lay S, PNAS 1997*.

Thank you for highlighting this excellent example of the less-is-more effect. We were familiar with it from the Ouchterlony double diffusion experiments of our youth. However, we had quite overlooked its connection to our current work. In response we have deleted the last sentence of the Introduction and added a sentence to the end of the first paragraph of the Discussion, in which we cite the Bray and Lay paper.